# Patterns of genomic deletions in monkeypox virus during the 2022 outbreak in the United States

Crystal M. Gigante [1], Daisy McGrath [1], Sophia Sukkestad[1], Hui Zhao[1], Mengfei Peng[1,2], Jade Takakuwa[1], Kimberly Wilkins[1], Vaughn Wicker [1], Audrey Matheny [1], Theodora Khan[1], Whitni B. Davidson[1], Mili Sheth[1], Alex Burgin[1], Julie Hand[3], Gillian Richardson[3], Danielle Haydel[3], Mark Burroughs[1], Jasmine Padilla[1], Justin S. Lee [1], Dhwani Batra[1], Ethan E. Hetrick [1], Dakota T. Howard[1,4], Kyle O'Connell[1,4], Jessica L. Rowell[1,4], Shatavia Morrison[1,4], Sarah Gillani[5], Michelle Lee[5], Anil Mangla[5], David Blythe[6], Robert Myers[6], Lynsey Kovar[1,7], Matthew H. Seabolt [1,7], Michael R. Weigand [8], Panayampalli S. Satheshkumar [1], Andrea M. McCollum[1], Christina L. Hutson[1] & Yu Li[1]

Poxviruses have a low overall rate of point mutations but are known to exhibit genomic duplications and deletions that can influence viral evolution. We examine the prevalence of large deletions in clade IIb monkeypox virus (MPXV) genomes during the global 2022 outbreak. We observe thirty-one distinct deletions, ranging from 573 to 21,576 bp among over 2000 MPXV genomes during 2022 – 2023 in the United States (U.S.). Almost all deletions are present in the first 25,000 bp or last 50,000 bp of the MPXV genome, excluding the terminal 500 bp. The large deletions result in extensive predicted gene loss as well as novel predicted gene products. Most unique deletions are observed in one case; however, one 3370 bp deletion mutant predominated in a U.S. state during late 2022 and a different 913 bp deletion may have arisen independently multiple times across several MPXV sublineages and multiple countries. The recurrent presence of large deletion mutants provides evidence of a mechanism of poxvirus evolution by genomic deletion and gene loss. While no deletion emerged in a dominant variant during 2022 – 2023, large deletions have the potential to result in viruses in which a therapeutic or diagnostic target is deleted.

Viral zoonoses represent continued threats to public health, causing several extensive, global outbreaks over the past decades. Understanding how zoonotic viruses change during extended transmission in humans is important in understanding viral adaptation to a new host. Mpox (previously called monkeypox) has gained international attention during the past two years following a global outbreak starting in 2022 and an ongoing multi-national outbreak in 2024. Mpox (caused by monkeypox virus, MPXV) was recognized as a potential

[1]National Center for Emerging and Zoonotic Infectious Diseases, Centers for Disease Control and Prevention, Atlanta, GA, USA. [2]Association of Public Health Laboratories, Bethesda, MA, USA. [3]Louisiana Department of Health, New Orleans, LA, USA. [4]Biomedical Data Science, Deloitte Consulting LLP, Arlington, VA, USA. [5]DC Health, Washington, D.C., USA. [6]Maryland Department of Health, Baltimore, MD, USA. [7]Public Health Office, Leidos Inc., Reston, VA, USA. [8]Office of Laboratory Systems and Response, Centers for Disease Control & Prevention, Atlanta, GA, USA. ✉e-mail: lzu1@cdc.gov

epidemic threat years prior to these outbreaks due to waning immunity after routine smallpox vaccination was ceased and re-emergence of cases in Africa[1,2].

MPXV is a member of the genus *Orthopoxvirus* in the family *Poxviridae*. Orthopoxviruses are zoonotic large double-stranded DNA viruses that have a broad host range across mammals, and several species can infect humans[3]. Among orthopoxviruses, certain species or strains cause more severe disease and mortality in some hosts than others. Variola virus (the causative agent of smallpox) had a narrow host range restricted to humans and caused severe disease and the highest human mortality among orthopoxviruses[4]. MPXV has a broad host range and can cause human disease similar to smallpox with lower mortality, ranging from <0.1% to over 10%, depending on the study[5]. Historical cases of mpox involved known or suspected zoonotic introduction into humans with limited spread within a household or among close contacts[6,7]. Recent mpox outbreaks have involved extensive human-to-human spread with no zoonotic intermediate[8–10], and persistent mpox cases continue in the human population across many countries after global expansion in 2022[11,12]. It is critical to monitor viral changes during expansion and potential establishment in a new host.

Viruses use a variety of mechanisms to generate genetic diversity. MPXV has a large linear double-stranded DNA genome of approximately 200 kb with a conserved central region containing genes involved in essential functions and less conserved genomic termini containing non-essential genes involved in immune modulation, host range, virulence, and unknown functions[13]. In general, DNA viruses accumulate single-nucleotide changes (SNPs) slower than RNA viruses due to higher fidelity replication machinery. Orthopoxviruses, including MPXV, have an estimated mutation rate of approximately 0 – 2 SNPs per genome per year[9,14,15]. However, recent reports have found increased SNP mutation rates in MPXV during extensive human-to-human transmission due to mutations induced by the APOBEC3 host antiviral protein family[8–10,16]. Still, the increased SNP mutation rate observed in MPXV during recent outbreaks is below that of many RNA viruses, at approximately 6 APOBEC3 signature mutations per genome per year[9].

Poxviruses compensate for a relatively low rate of point mutations by exhibiting genomic duplications, losses, and gains by recombination and horizontal gene transfer[17–21]. SNP differences do account for important variation between poxvirus species and isolates, and even single SNPs can lead to phenotypes such as drug resistance[22–24]. However, large genomic changes generate rapid, sometimes drastic diversity in viral populations, including deletions and duplications of genes. Deletions, gene duplications, and genomic rearrangements in poxvirus genomes have been associated with differences in pathogenicity and host range of different viruses[17,25–28]. Gene content is thought to correlate with host range in orthopoxviruses since zoonotic orthopoxviruses such as cowpox viruses and MPXV contain more intact coding sequences than host-restricted viruses, such as variola virus[29]. This correlation has led to the theory that a reduction in gene content plays an important role in orthopoxvirus speciation and host species restriction[18,19,29].

Differences in genome content can be observed between poxvirus genomes from different species, different clades, or in individual isolates. Genomic deletions, insertions, and duplications that occurred during poxvirus evolution have been inferred based on comparative genomic studies[18,19,27,30,31]. Five large insertions - deletions separate the major (Clade I from Clade II) and minor (Clade IIa and IIb) clades of MPXV, altering the number of predicted genes between clades by spanning entire open reading frames (ORFs) or disrupting start and stop codons[30]. Similar large insertions and deletions exist between variola virus major and minor clades, which also impact the presence/absence of predicted genes. In addition to established insertions and

deletions shared among species or viral clades, some large deletions have only been detected in a single virus.

Genomic duplications and deletions that affect viral pathogenicity have been observed in cultured vaccinia virus (VACV) under conditions that promote selection, leading to the development of vaccine virus strains and the proposed 'genomic accordion' mechanism of evolution for poxviruses[27,32]. Deletion of virulence and host range genes in MPXV can lead to viral attenuation[33] and have been hypothesized to underlie the differences in disease severity and case fatality between Clade I and Clade II mpox[34,35]. Recent genomic deletions in MPXV strains have resulted in direct impacts to public health, including drug-resistant viruses as well as viruses that could not be detected using specific real-time PCR tests[36–38]. The prevalence of genomic deletions in poxviruses under different evolutionary pressures, such as within the reservoir species compared to during population expansion in a new host (i.e., during an outbreak involving human-to-human transmission), is not well understood.

During 2022–2023, an international effort was made to characterize MPXV strains causing a global outbreak, track virus evolution, identify emerging lineages, and detect mutations of concern. While many studies have reported on SNP changes in MPXV sequences, the outbreak represents an unprecedented opportunity to investigate alternative, less frequent mechanisms of poxvirus mutation during natural infection and transmission in a new host. In this study, we describe genomic surveillance of large deletions in MPXV genomes in the U.S. during 2022.

## Results

We examined MPXV genome sequences produced from U.S. mpox cases collected during 2022 at the Centers for Disease Control and Prevention (CDC) for the presence of large deletions. Insertions, deletions in regions of repetitive sequence, and more complex genomic rearrangements were excluded from this analysis. Thirty-one unique deletions greater than 500 bp were detected in 64 samples out of 2362 Clade IIb lineage B.1 MXPV genomes sequenced during routine surveillance at CDC in the U.S. during 2022–2023 (-2.7%). Deletion sizes ranged from 573 to 21,576 bp (approximately 3 to 11% of the MPXV genome) based on comparison to Clade IIb lineage B.1 reference genome 2022-USA-MA001 (ON563414.3) (Table 1, Data S1, Data S2). Large deletions could be easily visualized as a gap in sequencing read coverage mapped to reference 2022-USA-MA001 (Fig. 1). Deletion boundaries were confirmed by visualization of even read mapping across the deletion in draft genomes and presence of reads spanning the deletion (Fig. 1, Fig. S1, Supplementary Methods).

In general, deletions were located towards the terminal ends of the MPXV genome (Fig. 2). Except for one 640 bp deletion (MPXVdel 3, Table 1, Fig. 2), all deletions were located within the left 25,000 or right 50,000 bp of the genome. The 640 bp deletion occurred around position 135,000 (approximately 62,000 bp from the right genomic terminus) and was predicted to impact p4c precursor, which is a major component of IMV surface tubules. Eleven deletions spanned tandem repeat regions, resulting in loss of these low complexity regions from the genome: three spanned the longest AT repeat, one spanned the 9mer CATTATATA repeat, and seven spanned the 16 mer CATAAGTTAGTTAAGT repeat located in the ITRs. Several deletions extended into the inverted terminal repeats (ITRs). Two deletions (MPXVdel 2 and 7) were confined within the ITRs and thus were duplicated on either end of the genome (Fig. 2, Table 1). The remaining ITR-spanning mutations only affected one terminus, thus shortening the size of the homologous sequence found in both termini (ITR size) and making several genes that are typically diploid in MPXV genome haploid in the deletion virus. Sixteen unique deletions resulted in ITR shortening; the shortest ITR length was observed in one 8534 bp deletion that produced a genome with 447 bp ITRs (MPXVdel 19, Table 1). MPXV genomes with deletions affecting the ITRs could not be

**Table 1 | Unique deletions observed in CDC sequence surveillance of Lineage B.1 MPXV from the U.S. in 2022**

| No. | Size# | Start# | End# | Cases^ | Affected CDS^^ |
|---|---|---|---|---|---|
| 1 | 573 | 181,836 | 182,408 | 1 | MPXVgp182: surface glycoprotein B21 |
| 2 | 600 | 2159 | 2758 | 1* | MPXVgp002: TNF receptor CrmB (2 copies) |
| | | 194,446 | 195,045 | | |
| 3 | 640 | 135,539 | 136,178 | 1 | MPXVgp137 (COP-A25L): A-type inclusion protein |
| | | | | | MPXVgp138 (COP-A26L): IMV surface tubules major component |
| 4 | 913 | 11,343 | 12,255 | 12* | MPXVgp010 (OPG023): ankyrin-like host range protein |
| 5 | 1076 | 149,053 | 150,128 | 1 | MPXVgp155 (COP-A44L): 3 beta-hydroxysteroid dehydrogenase/delta 5- > 4 isomerase |
| | | | | | MPXVgp156 (COP-A45R): virion core protein |
| 6 | 1120 | 8209 | 9328 | 1 | MPXVgp007: IL-1 receptor antagonist |
| 7 | 1192 | 1571 | 2762 | 1 | MPXVgp001 (J1L/R) (2 copies): chemokine-binding protein |
| | | 194,444 | 195,635 | | MPXVgp002: TNF receptor CrmB (2 copies) |
| 8 | 1498 | 8211 | 9708 | 1 | MPXVgp007: IL-1 receptor antagonist; MPXVgp008: zinc finger-like virulence factor |
| 9 | 2350 | 176,519 | 178,868 | 2 | MPXVgp178 (COP-B20R): Ankyrin-like protein |
| 10 | 2475 | 155,787 | 158,261 | 1 | Intergenic |
| 11 | 2635 | 157,206 | 159,840 | 1 | MPXVgp162 (COP-A55R): BTB Kelch-domain containing protein CRL complex |
| | | | | | MPXVgp163 (COP-A56R): EEV envelope and cell membrane glycoprotein |
| 12 | 2668 | 20,157 | 22,843 | 1 | MPXVgp023 (COP-N1L): secreted virulence factor; MPXVgp024 (COP-N2L): alpha amanatin target |
| | | | | | MPXVgp025 (COP-M1L): ankyrin-like; MPXVgp026 (COP-M2L): NFkB inhibitor |
| 13 | 2727 | 12,546 | 15,272 | 1 | MPXVgp010 (OPG023): ankyrin-like host range protein |
| | | | | | MPXVgp011: ANK-containing protein; MPXVgp012: Type-I interferon resistance |
| 14 | 3302 | 188,069 | 191,370 | 1 | MPXVgp185: Toll-like receptor signaling inhibitor, Bcl-2 like |
| | | | | | MPXVg187; MPXVgp188: ankyrin-like protein |
| 15 | 3370 | 155,340 | 158,709 | 28** | MPXVgp161 (COP-A51R); MPXVgp162 (COP-A55R) |
| 16 | 3493 | 173,232 | 176,724 | 2 | MPXVgp177 (COP-B19R); MPXVgp178 (COP-B20R) |
| 17 | 3852 | 170,150 | 174,001 | 1 | MPXVgp172 – 175 |
| 18 | 4430 | 4222 | 8651 | 1 | MPXVgp003 – 006 |
| 19 | 6469 | 189,436 | 195,504 | 1 | MPXVgp185, 187 – 191 |
| 20 | 8207 | 2166 | 10,372 | 2 | MPXVgp002 – 008 |
| 21 | 8534 | 447 | 8980 | 1 | MPXVgp001 – 006 |
| 22 | 8745 | 14,958 | 23,702 | 1 | MPXVgp012 – 016 |
| 23 | 10,727 | 181,067 | 191,793 | 2 | MPXVgp181 – 182; 185, 187, 188 |
| 24 | 11,161 | 180,698 | 191,858 | 2 | MPXVgp181, 182, 185, 187, 188 |
| 25 | 11,269 | 181,482 | 192,750 | 1 | MPXVgp182(surface glycoprotein B21); 185, 187 – 189 |
| 26 | 12,664 | 173,168 | 185,831 | 1 | MPXVgp175; 177 – 182(surface glycoprotein B21) |
| 27 | 14,230 | 179,405 | 193,634 | 1 | MPXVgp180 – 182; 187 – 189 |
| 28 | 14,410 | 182,032 | 196,441 | 1 | MPXVgp182(surface glycoprotein B21); 185, 187 – 191 |
| 29 | 16,208 | 175,565 | 191,722 | 1 | MPXVgp181, 182, 185, 187 – 191 |
| 30 | 16,415 | 178,827 | 195,241 | 1 | MPXVgp180 – 182, 185, 187 – 190 |
| 31 | 21,576 | 1230 | 22,805 | 1 | MPXVgp001 – 016; MPXVgp021 –26 |

#Deletion size (bp) and start and end coordinates are relative to ON563414.3.

^Number of associated mpox cases

^^Predicted changes to coding sequences. Gene names are based on gene orthologs in the MPXV genome (designated MPXVgp) or Vaccinia virus Copenhagen (COP-). Additional details can be found in Table S1.

*Additional sequences with mutation found in public databases.

**Only six sequences sharing this deletion were observed through routine surveillance. An additional 22 sequences were observed by selectively sequencing and obtaining additional samples from Louisiana.

No.: Deletion number.

assembled using graph reconstruction feature of PolkaPox (https://github.com/CDCgov/polkapox) and were manually assembled.

We next examined the predicted impact of each of the 31 unique deletions on predicted genes. The impact ranged from no effect (for one intergenic deletion) to a 21,576 bp deletion that impacted 22 predicted coding sequences (CDS) (Table 1, Supplementary Table 1). The genomic positions of several deletions overlapped (Fig. 2), and so the predicted CDS were the same. Many genes impacted had predicted roles in host range or host immune interaction, which is expected for genes located in the genomic termini. Predicted truncations and novel

fusion proteins involving the monkeypox-specific major antigenic surface glycoprotein B21 encoded by MPXVgp182 (OPG210) were observed in over one-third (11/31) of the unique deletions observed in this study. Deletions affecting known host range genes were observed in 12 of the unique deletions, including MPXVgp180(SPI-1) in 3 of the 31 unique deletions, MPXVgp027 (COP-K1L) in one deletion, MPXVgp013 (COP-C7L) in 2 of 31 deletions, MPXVgp010 (CP77) in 3 of 31 deletions, and MPXVgp008 (p28) in 3 of the 31 unique deletions. TNF receptor-like (CrmB) homolog MPXVgp002/190(OPG002) was disrupted in 9 of the 31 unique deletions, although in some cases only one copy was

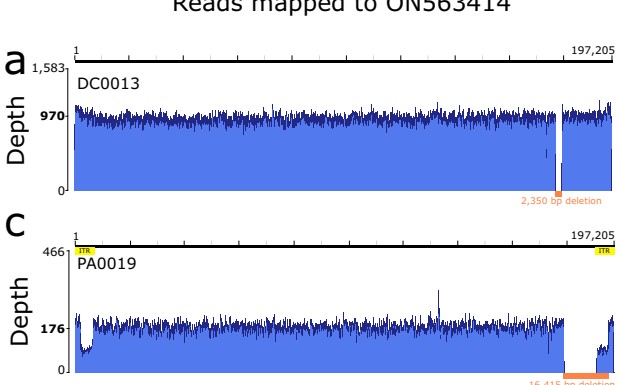

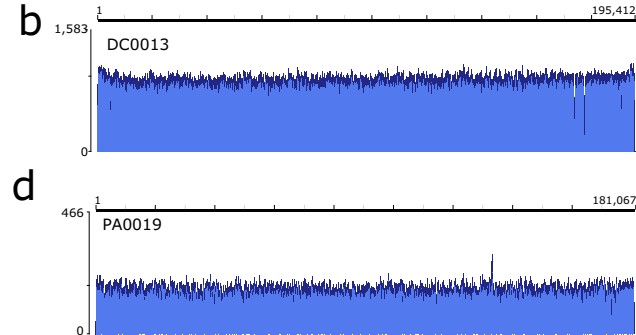

**Fig. 1 | Read mapping profiles from metagenomic sequencing of MPXV genomes from two samples with large deletions.** Reads mapped to reference sequence ON563414.3 show a gap or decrease in coverage over deletion regions (highlighted in orange with size of deletion in orange text) (**a**, **c**). **b** Even read coverage profiles produced when reads were mapped to final genomes containing deletions (**b**, **d**). Read depth is shown on the y axis; mean read depth across the genome is shown in bold text. Genomic position is shown above each graph; tick marks are approximately every 10,000 bases. Additional read mapping profiles for samples with low average read depth can be found in Fig. S2.

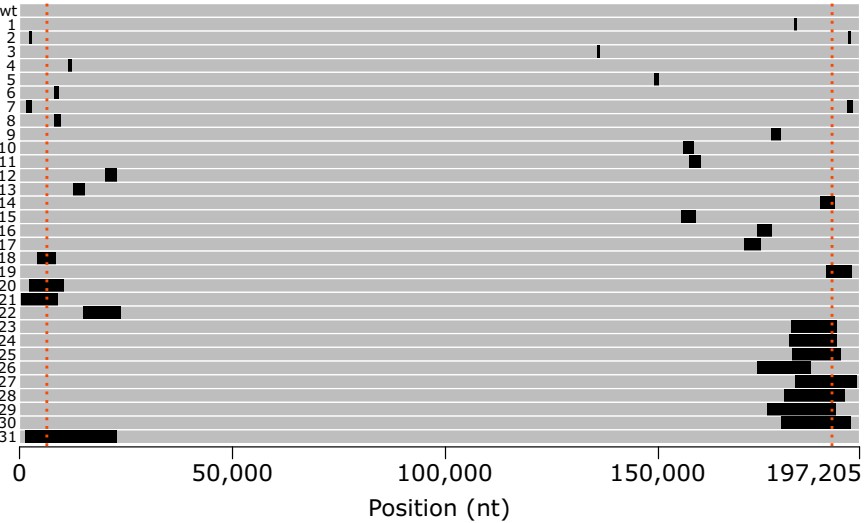

**Fig. 2 | Relative locations of 31 unique genomic deletions >500 bp in MPXV Lineage B.1 genomes.** Deletion locations are shown in black on an MPXV genome depicted as a gray bar. Position is based on ON563414.3 (wt, no deletions). Each unique deletion is depicted on a single genome with numbers corresponding to index numbers in Table 1. Order is the same as in Table 1 with the smallest deletion at the top and the largest at the bottom. Red dotted lines indicate the boundaries of the inverted terminal repeats (ITRs).

affected, leaving the second copy intact. Four ankyrin-repeat domain-containing homologs were disrupted in the deletion MPXVs, including MPXVgp004 (Cowpox virus Brighton Red 017) in 4 of 31 deletions, MPXVgp010 (CP77) in 3 of 31 deletions, and MPXVgp012 in 3 of 31 deletions. Of note, several deletions resulted in the potential formation of novel proteins, including two small in-frame deletions, several predicted C- and N-terminal truncations, and six potential fusion products when deletions spanned from the CDS of one gene in-frame to the CDS of another. It is likely that even though these CDS continued in frame, they would likely produce non-functional or misfolded products. Details can be found in Data S1.

Most large deletions were only observed in a single MPXV genome; however, eight deletions were observed in two or more cases (Table 1). Five deletions were found in two cases (Table 2). For these five deletions, virus genomes that shared the same exact deletions differed by 2–8 SNPs (average; 4.6). Investigation of a 2350 bp deletion revealed that the patients associated with these MPXV sequences were long-term partners with a common sexual encounter with a confirmed mpox case in Europe. Two SNPs were observed between the whole

genome sequences associated with those cases (MD0005 and DC0013). Case investigations were not made into the remaining four deletions (each found in two cases); however, for three of the four unique deletions, both cases sharing the deletion were from the same state with similar collection dates (Table 2).

Three deletions were observed in more than 2 cases. One 3370 bp deletion (MPXVdel 14 in Table 1, Fig. 2) was detected in 28 MPXV sequences. This deletion spanned the C-terminus of VACV-COP A51R ortholog CDS (OPG181, poxvirus B1R BTB Kelch-domain containing protein CRL complex), intergenic sequence, and into the N-terminus of VACV-COP A55R ortholog CDS (MPXVgp162, poxvirus B1R BTB Kelch-domain containing protein CRL complex). The deletion did not produce a frame shift and continuing the reading frame from COP-A51R resulted in a predicted fusion protein containing the N-terminal 312 amino acids from the COP-A51R ortholog followed by the C-terminal 49 amino acids from the COP-A55R ortholog (Fig. S3). The 28 MPXV sequences containing this deletion were generated from samples from 25 cases from Louisiana and 1 each from Utah, Texas, and Georgia. The cases in Utah and Texas both had confirmed travel and sexual

encounters in Louisiana prior to illness onset. The deletion was first detected in a case from July 2022 and was detected in a majority of MPXV from Louisiana during November and December 2022, although only seven samples were tested (Fig. 3a, Data S3). Phylogenetic analysis revealed the MPXV sequences containing the 3370 bp deletion formed a monophyletic clade, separated from other MPXV sequences from Louisiana, suggesting a common ancestor (Fig. 3). Constraining the sequences containing the 3370 bp deletion to be monophyletic produced a tree with similar log likelihood as the unconstrained tree showed in Fig. 3 (−258498.6486 vs −258498.6487, $p = 0.419$ and 0.678 by Expected Likelihood Weight[39] and approximately unbiased (AU)[40]

**Table 2 | Details for five pairs of cases associated with MPXV sharing the exact same deletion**

| MPXVdel | Collection Date# | State^ | Deletion Size^^ | SNPs* |
|---|---|---|---|---|
| 9 A | 2022-06-23 | DC | 2350 | 2 |
| 9B | 2022-06-23 | MD | 2350 | |
| 16 A | 2022-07-26 | NC | 3493 | 8 |
| 16B | 2022-07-12 | OH | 3493 | |
| 20 A | 2022-08-21 | CA | 8207 | 7 |
| 20B | 2022-08-07 | CA | 8207 | |
| 23 A | 2022-11-10 | TX | 10,727 | 2 |
| 23B | 2022-11-05 | TX | 10,727 | |
| 24 A | 2022-08-29 | NC | 11,161 | 4 |
| 24 B | 2022-08-30 | NC | 11,161 | |

#Collection date is in YYYY-MM-DD format.
^DC District of Columbia, MD Maryland, NC North Carolina, OH Ohio, CA California, TX Texas. Additional details can be found in Table S1.
^^Deletion size (bp) is relative to ON563414.3.
*SNPs Single nucleotide polymorphisms; do not include differences due to ambiguous bases or regions of low complexity.

test, respectively). Together with the epidemiological data, the phylogenetic analysis suggests MPXV sequences with the 3370 bp deletion were derived from a common ancestor and spread locally in Louisiana, with few travel-related cases in other states. Comparison of case histories for a subset of cases associated with the 3370 bp deletion revealed no striking differences in disease severity and clinical outcomes compared to other MPXV cases in Louisiana during the same time.

A 913 bp deletion (MPXVdel 4, Table 1, Fig. 2) was detected in 12 MPXV genomes during routine surveillance at CDC. An additional 49 MPXV genomes were found in public databases that likely contained the same deletion (either had gap or Ns in spanning the 913 bp deletion) (Data S4, Figs. 4, S4). Sequences were from several U.S. states, Portugal, Germany, and Poland (Data S4). MPXV sequences were produced by at least four different laboratories, some of which used tiled amplicon approaches while others used shotgun metagenomics. The 913 bp deletion was confirmed by PCR (Fig. S5). The 913 bp deletion resulted in a C-terminal truncation of the putative host range OPG023 (MPXVgp010; ankyrin/host range, Bang-D8L) ortholog, resulting in a 225 amino acid protein ending in a novel four amino acids due to a frame shift caused by the deletion. The predicted wild-type protein is 660 amino acids. MPXV genomes with the 913 bp deletion were assigned to multiple sublineages of Clade IIb, including B.1.2, B.1.3, B.1.9, and B.1.15, by Nextclade (Data S4). Phylogenetic analysis of sequences containing the mutation were dispersed among several B.1 sublineages (Fig. 4). Constraining the sequences with the 913 bp deletion to be monophyletic produced a tree with lower log likelihood compared to the unconstrained tree [−253193.0215 vs −253018.451, $p = 2.54e-31$ and 3.35e-76 by Expected Likelihood Weight and approximately unbiased (AU) tests, respectively], suggesting the data are better fit to the unconstrained phylogeny. Together, the phylogenetic analyses suggest this deletion may have arisen independently and/or reverted multiple times during the 2022 outbreak.

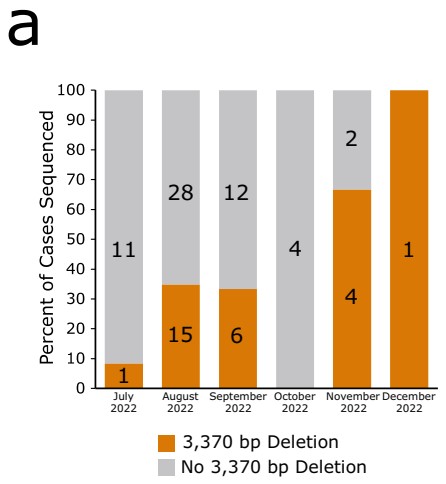
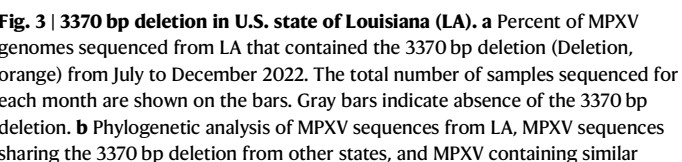
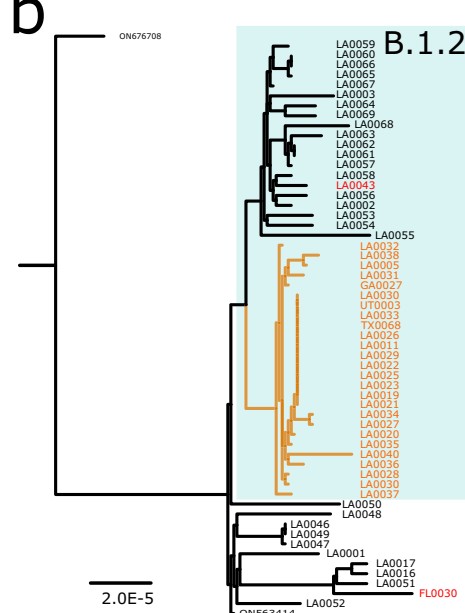

**Fig. 3 | 3370 bp deletion in U.S. state of Louisiana (LA). a** Percent of MPXV genomes sequenced from LA that contained the 3370 bp deletion (Deletion, orange) from July to December 2022. The total number of samples sequenced for each month are shown on the bars. Gray bars indicate absence of the 3370 bp deletion. **b** Phylogenetic analysis of MPXV sequences from LA, MPXV sequences sharing the 3370 bp deletion from other states, and MPXV containing similar deletions. Deletion sequences formed a monophyletic clade (orange branches) separate from LA sequences with no deletion (black branches) within MPXV clade IIb lineage B.1.3. Sequences are labeled with isolate names listed in Data S1 and S2 or GenBank accession numbers for reference sequences. US states are abbreviated as follows, LA Louisiana, FL Florida, GA Georgia, UT Utah, TX Texas. Scale bar indicates substitutions per site. Additional sample details can be found in Data S3.

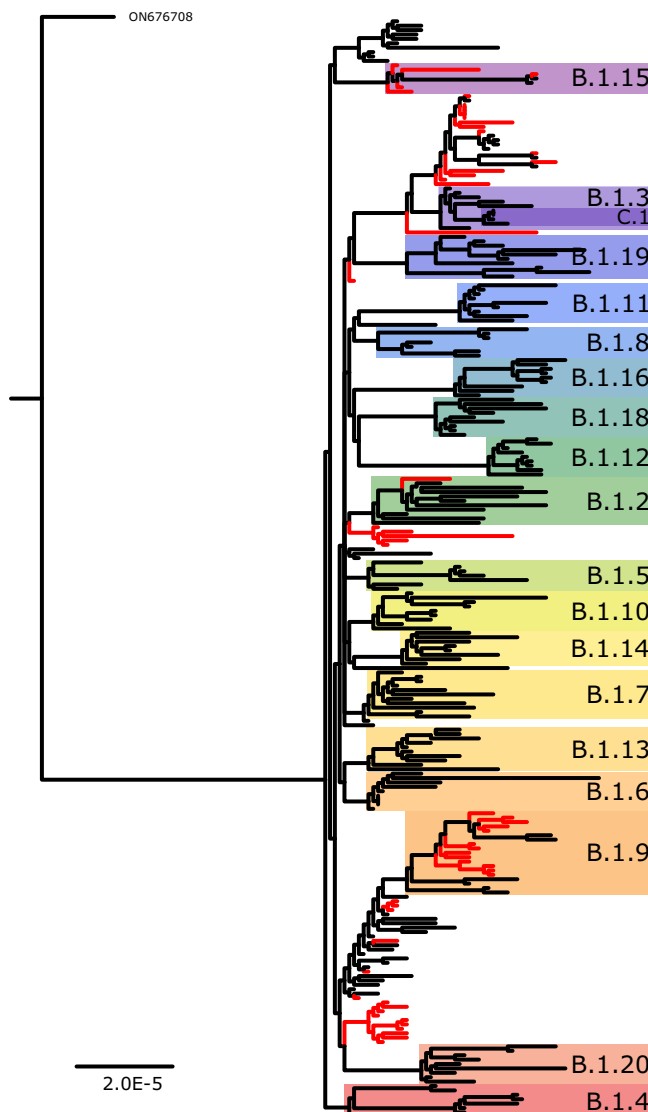

**Fig. 4 | Phylogenetic analysis of global MPXV genomes sharing the 913 bp deletion and representative MPXV sequences.** MPXV genomes containing the 913 bp deletion are highlighted by red branches while sequences with no deletion are shown by black branches. Colored boxes indicate lineage assignment using NextClade[73]. Scale bar indicates substitutions per site. A duplicate tree with accession numbers can be found in Fig. S4. Additional details for samples can be found in Data S1 and Data S4.

Lastly, a 600 bp deletion (MPXVdel 2) observed in MPXV sequences from a cluster of 12 cases in California, USA[37] was observed in this study in MPXV from New York City. This 600 bp deletion cluster and the 1192 bp deletion (MPXVdel 7) represent mutations of concern because deletions occur in the target regions of an MPXV generic and a Clade II-specific mpox diagnostic PCR test[41]; and are not detected by the clade II-specific test (Data S2, samples were not tested by the MPXV generic test)[37]. However, there has been no evidence of recent spread of MPXV strains containing either deletion.

## Discussion
We report large deletions among a large number of MPXV genomes during the 2022 outbreak. Smaller scale studies have also reported large deletions in MPXV during the 2022 global outbreak, including two large deletions among 56 mpox cases (3.6%) in Australia[42] and three large deletions among 207 mpox cases (1.4%) from the U.S. states of Washington and Ohio[43]. Prior to 2022, large genomic deletions have

been seen in variola virus, cowpox virus, VACV, and all clades of MPXV[25,30,38,44–46]. One such case was found in a Clade IIb MPXV isolate from Bayelsa, Nigeria in 2018 (GenBank MT903341)[46]. However, the limited number of total sequences makes it difficult to determine if the increased reports of large deletions in MPXV during the 2022 global outbreak are due to increased sequence surveillance or a true increase in the occurrence of deletions in recent years. Other reports have described geographic clusters of mpox cases caused by MPXV sharing the same, small genomic deletion[36,37]. Determining the prevalence of large deletions from public sequence data is challenging because often large spans of undetermined bases (Ns) may indicate low coverage, amplicon drop-out, or a potential deletion. Correct identification of genomic deletions may be influenced by sequencing approach, as deletions are more easily detected using a direct metagenomics sequencing approach compared to amplicon-based approaches, where deletions may be misinterpreted as amplicon drop-out. These changes may provide no advantage or even be deleterious to the virus, yet some changes may prove beneficial and/or persist by chance and play an important role in viral evolution. Accurate identification of deletions, insertions, or recombination events can be technically challenging, especially when using enrichment protocols that rely on PCR or probe hybridization, which can have inherent bias toward a wild-type virus. Large genomic changes or regions of low homology with the virus used to design the enrichment scheme may not be enriched or amplified, resulting in genomic regions of low coverage that regions may be filled with Ns and interpreted as amplicon drop-out or probe failure instead of investigated.

Previous studies have suggested that large deletions and rearrangements may involve imperfect base pairing with tandem repeat regions[26,28,47–50]. The deletions observed here differ from the mechanism of repeat expansion and contraction in low complexity regions described in other studies[51]. In our data, we did not observe deletion breakpoints within long tandem repeats; however, several deletions began or ended less than one kb from a tandem repeat or near a short adenosine homopolymer. Second, even though several of the deletion regions overlapped and involved many of the same CDS, most of the exact breakpoints were unique, suggesting they may be random. Further investigations are needed to understand the mechanism by which large deletions occur are needed.

Each deletion mutant was identified as the major or only virus in the clinical sample, suggesting the virus was replicative and none of the deleted genes were essential. Many of the predicted disrupted genes were associated with poxvirus virulence and host range (Data S3)[52,53]. We observed deletions affecting several known host range genes, including COP-K1L and SPI-1, where disruption is predicted to be involved in limiting host range. K1L has been independently disrupted in three orthopoxvirus lineages with narrow host ranges: variola virus, camelpox virus, and taterapoxvirus[53], and its deletion in VACV results in reduced growth in human cells[54]. Deletion of SPI-1 from rabbitpox virus (but not cowpox virus) also prevents growth in human cells[55]. Several of the genes that were lost in MPXV deletion viruses have also been shown to cause viral attenuation and/or the ability to grow in specific cell lines when lost in targeted or large deletions in VACV strains[56,57]. Four ankyrin-repeat domain-containing homologues present in MPXV that are disrupted in variola virus[52,53] were disrupted in the deletion MPXVs, including MPXVgp004 (Cowpox virus Brighton Red 017) in 4 of 31 deletions, MPXVgp010 (CP77) in 3/31 deletions, and MPXVgp012 in 3/31 deletions. Predicting the impact of the large deletions we observed that involve multiple CDS is difficult, and further in vitro and animal studies are needed to better understand how large deletions affect viral pathogenesis and host range. However, previous research in other OPXVs suggests deletion or disruption of homologs of the genes that were affected across several deletion-containing MPXV strains in this study may result in altered host range or virulence[52,58–62], which could affect

transmissibility in humans; however,r there is a lack of needed studies in animal models and in MPXV. Persistence or emergence of a deletion virus could contribute to the adaptation of MPXV to humans or its ability to infect non-human mammals.

Three large deletions were detected in more than 1–2 cases. A 3370 bp deletion observed in several dozen cases from Louisiana involved a potential fusion of two BTB Kelch-domain containing proteins (VACV-COP A51R ortholog with an A55R-like C-terminus). Kelch-domain containing proteins are generally thought to be non-essential and are hypothesized to impact host range, viral adaptation to host(s) or in vivo species-specific properties[45,63,64]. It is difficult to predict the impact this fusion protein would have on the virus since Kelch-domain containing proteins are present in numerous copies in orthopoxvirus genomes, in variable lengths. The earliest samples of the deletion virus were from Utah and Texas, both in travelers to Louisiana. It is currently unclear whether the deletion occurred to a virus in Louisiana or was introduced by a traveler, but almost all subsequent viruses with the 3370 bp deletion were from Louisiana, where it likely spread by chance to become a major variant. Limited clinical investigation revealed no evidence of increased severity of cases caused by viruses containing the 3370 bp deletion. We do not have evidence for why this 3370 bp deletion would convey a selective advantage, but it did not seem to convey a negative transmission phenotype.

A 913 bp deletion was observed in 12 MPXV genomes sequenced at CDC and 45 additional sequences from other laboratories. The deletion results in truncation of the predicted host range ankyrin repeat protein encoded by the OPG023 ortholog, including deletion of the ankyrin repeat and PRANC/Fbox domains. This deletion has been reported in one study describing ORF-disrupting mutations in MPXV genomes in the U.S. state of Washington[43]. OPG023 was identified as one of ten proteins more prone to mutation in MPXV during the 2022 outbreak[65]. It is difficult to predict the effect of OPG023 truncation on the virus, since ankyrin-domain-containing proteins are numerous in MPXV genomes. Additional characterization of the virus containing the 913 bp deletion is warranted, as this deletion has arisen in multiple B.1 sublineages, which could indicate some advantage or adaptation in that virus. Alternatively, it could also be a disadvantageous deletion that has reverted. Further investigation may provide insight into the mechanism behind poxvirus genomic deletions if some genomic feature is increasing the likelihood of deletion at that location.

Previous studies have revealed that poxvirus ITRs are hot spots for large deletions and duplications[26,28,47–50], and several of the large deletions observed in this study resulted in shortened ITRs to less than 1000 bp. No deletions extended through or eliminated the ITRs or involved the concatemer resolution sequence, as expected, since they are required for poxvirus replication[66,67]. The shortest ITR length observed in this study was close to 500 bp (although genomes were not sequenced to the hairpin termini) and contained no genes, which is similar to ITRs seen in variola virus. Among orthopoxviruses, variola virus genomes have the shortest ITRs (200–500 bp, containing no genes), and VACV ITRs can be larger than 10 kb and contain six diploid genes[29].

The concentration of large deletions in the first 25 kb and last 50 kb of the MPXV genome is informative for public health in the design of diagnostic tests and therapeutics. Deletions in regions used in diagnostic tests can lead to false negative results, allowing for unrecognized spread of the pathogen. For poxviruses, deletions affecting the performance of diagnostic tests have been seen in cowpox virus[44] and MPXV[37,38]. In this study, we observed deletions affecting the target of published Clade II-specific and MPXV generic PCR assays (J2R/L, CrmB gene, Tumor necrosis factor (TNF) receptor gene homolog)[41] in 9 of the 31 unique deletions; however, only two of the unique deletions deleted the assay target sites in both gene copies. One of these deletions was associated with a cluster of cases in California in 2022 but has not been detected in over 1000 MPXV genomes analyzed since 2023[37]. These types of deletions are not limited to Clade

II, as a deletion in Clade Ib MPXV associated with a multinational outbreak in Africa precludes detection by a Clade I-specific real-time PCR assay (Complement Control Protein gene C3)[10,38,41], potentially thwarting diagnostic testing in the midst of a large scale outbreak if the diagnostic laboratories had not been using approaches to target multiple genomic regions including a viral conserved OPXV gene[10,38,68]. Diagnostic tests or medical countermeasures with targets located in the center of the genome, especially in an essential gene and a critical functional domain, are less likely to be impacted by mutations. Several PCR assays target the DNA polymerase gene (VACV-COP E9L), including the CDC FDA cleared non-variola orthopoxvirus (NVO) test (E9L-NVAR in the original publication) and the orthopoxvirus laboratory-developed test (OPX3)[68,69]. A mpox generic test targeting the double-stranded RNA binding protein (VACV-COP E3L ortholog) is also located within the central, conserved region[70]. Using an assay that targets an essential gene or using multiple assays that target different regions of the genome may be necessary to avoid false negative results for viruses with large deletions[68]. Likewise, large deletions have the potential to impact vaccines and therapeutics that impact single targets or few antigens.

This characterization of deletions in the MPXV genome may also be used to inform the design of new therapeutics. Deletions in targets of therapeutic drugs or vaccines can also cause major issues, especially when limited alternatives are available, as is the case for orthopoxvirus infections. None of the large deletions presented here involved the VACV-COP F13L gene ortholog, which encodes the target of tecovirmat (brand name TPOXX®, Siga Technologies), an antiviral drug used for the treatment of orthopoxvirus infections. However, smaller deletions and insertions in F13L are known to cause resistance to tecovirimat and have been observed in MPXV strains during 2022- 2023[22,23,36]. We found the monkeypox-specific major antigenic surface glycoprotein B21 encoded by MPXVgp182 (OPG210) was deleted, truncated, or otherwise disrupted in over one-third (11/31) of the unique deletions observed in this study. OPG210 was also identified as one of ten proteins prone to mutation during the 2022 outbreak[65]. This raises concern over the proposed use of B21 as a potential MPXV-specific target for serology studies[71]. Finding natural viruses with deletions in major antigenic proteins supports the use of vaccines and serological tests that target multiple virus proteins, preferably situated far away in the genome.

It has been hypothesized that gene duplication and deletion in poxviruses may help them evolve to new hosts or selection conditions[18,27,72]. MPXV strains described in this study arose during rapid expansion during an outbreak in a new host, which is expected to result in increases in genetic diversity. The majority of viral mutations observed in recent MPXV outbreak sequences have been attributed to host editing of the viral genome. The detection of 31 independent large deletions in this study implies large genomic deletions as a recurrent mechanism of generating genomic diversity in poxviruses during rapid population expansion, with the potential to impact viral evolution or escape medical countermeasures. The repeated observations of large deletions among MPXV strains during the outbreak indicate a continued need for genomic surveillance of MPXV and other orthopoxviruses to inform prudent approaches for diagnostics and medical countermeasures and ensure continued preparedness.

## Methods
### Sampling
MPXV DNA used in this study included remainders of clinical and non-clinical samples submitted to the Poxvirus Laboratory, Centers for Disease Control and Prevention (CDC) for routine testing. Viral sequencing of those samples for genomic surveillance was reviewed and deemed as non-research public health surveillance by the CDC. No specimen collection was performed for this study. Specimens from lesion swabs or crusts of patients were subjected to DNA extraction after inactivation with an EZ1 & 2 DNA tissue kit on an EZ1 Advanced XL

Instrument (Qiagen, www.qiagen.com). Real-time PCR was performed on the extracted DNA using an Applied Biosystems 7500 Fast Dx PCR instrument with a TaqMan fast advanced master mix (Thermo Fisher Scientific, www.thermofisher.com) using a clade II-specific qPCR assay[41]. Reactions were conducted in 20 μL volumes using thermocycling conditions 95 °C for 20 s followed by 40 cycles of 95 °C for 3 s, 63 °C for 30 s.

## Metagenomics Sequencing

Extracted DNA (15 μL) for all samples producing an average Ct <29 was used as input for the Illumina DNA Prep method according to the recommended protocol with ½ reagent volumes used throughout; samples were amplified using 5 or 11 PCR cycles depending on input concentration. Libraries were visualized using the Agilent Fragment Analyzer instrument and the HS NGS Fragment Kit (Agilent Technologies Inc., Santa Clara, CA). Forty-eight to 64 libraries were pooled at approximately equal molarity and sequenced (200 pM final loading concentration) on an Illumina NovaSeq 6000 instrument using the 300-cycle SP sequencing components.

Large genomic deletions were identified through visual screening of read mapping profiles for read mapping to the reference genome MPXV_USA_2022_MA001 (GenBank Accession ON563414.3). Potential deletion mutant sequences were identified by dips in read depth coverage. Deletion was confirmed by examination of raw read sequences at the boundaries of the suspected deletion. Additional details can be found in Supplementary Methods.

## Genome Assembly

MPXV genome sequences were assembled using PolkaPox v0.1-beta (https://github.com/CDCgov/polkapox). Briefly, Illumina sequencing reads were taxonomically classified as human or orthopox sequences using a Kraken2 v2.1.2 (https://github.com/DerrickWood/kraken2) with a custom database consisting of the MPXV genome MPXV_USA_2022_MA001 (GenBank Accession ON563414.3) and the human genome GRCh38 (GCF_000001405.40). MPXV-containing reads were extracted (seqtk v1.3, https://github.com/lh3/seqtk) and then trimmed, filtered, and adapters removed using FastP v0.23.2 (https://github.com/OpenGene/fastp). FastP parameters were set to filter out reads of PHRED quality less than 20 and lengths shorter than 50 base pairs. Genome contigs generated using Unicycler v0.4.8 (https://github.com/rrwick/Unicycler) were assembled into a draft genome using graph reconstruction from the Unicycler gfa output. The draft genomes were then polished by mapping reads back to the de novo assembly using bwa mem. Final assemblies were produced by consensus calling using IVAR. In cases where graph reconstruction failed, contigs were manually assembled. In all cases, the deletion junction site sequences were confirmed in raw reads as described in Supplementary Methods. Final read mapping to the draft genome was compared to read mapping to the reference genome MPXV_USA_2022_MA001 (GenBank Accession ON563414.3) to confirm assembly accuracy. For 15 genomes where the deletion overlapped with the ITRs, the graph reconstruction step of PolkaPox was unsuccessful at generating a whole-genome final assembly. For these cases, contigs were manually assembled in Geneious Prime before polishing as described above.

Genome annotation and submission to NCBI databases were performed using TOSTADAS. Nextclade assignment was performed using the Mpox virus All clades reference dataset updated at 2025-04-25 12:24:24 (UTC) using Nextclade v3.8.1[73] on clades.nextstrain.org. Raw sequence data and final assemblies are available on NCBI databases using the accessions listed in Table S2.

## Phylogenetic analysis

MPXV genomes were aligned using MAFFT v7.490 in Geneious Prime using FFT-NS-1. Alignment columns containing genomic termini, large deletions, and tandem repeat regions were manually removed. Final alignments included 252 sequences and 186,378 alignment columns for the 913 bp deletion and 60 sequences and 193,146 alignment columns for the LA 3370 bp deletion. Phylogeny was produced using IQ-TREE v 2.2.6[74] using model selection -MFP[75] and 1,000 bootstrap replicates using the -bnni flag. GenBank OP450997 (MPXV clade IIb, lineage A.2) and ON676708 (MPXV clade IIb, lineage A.1.1, USA MD 2021) were used to root the trees. A constrained tree was generated that forced monophyly of all sequences containing either the 913 bp or 3370 bp deletion using the newick format: "((deletion sequences list), control sequences list, outgroup);". Trees were then compared using IQ-TREE -n 0 -zb 10000, -au.

## Reporting summary

Further information on research design is available in the Nature Portfolio Reporting Summary linked to this article.

## Data availability

Assembled genomes and raw read data generated in this study were deposited to NCBI SRA and Genbank under accessions listed in Table S2. Accession numbers used in phylogenetic analysis are listed in Data S3. Alignments used for phylogenetic analyses are available in Data S5 and S6. MPXV genome sequences were assembled using PolkaPox v0.1-beta (https://github.com/CDCgov/polkapox). Genome annotation and submission to NCBI databases was performed using TOSTADAS.

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

## Acknowledgements

Nicholas Chen, Chantal Vogels, and the other members of the Grubaugh Lab in the Department of Epidemiology of Microbial Diseases, Yale School of Public Health, for the design of the MPXV tiled PCR-based amplicon protocol (Monkeypox virus multiplexed PCR amplicon sequencing (PrimalSeq) V.2. (protocols.io); technical contributions that were made by the staff of the Maryland Department of Health Laboratory; and efforts made by the 2022 Multi-National Monkeypox Outbreak Response teams at the Centers for Disease Control and Prevention, Florida Department of Health, DC Health. T. Khan, J. Takakuwa, V. Wicker, D. McGrath, and S. Sukkestad were supported in part by appointment to the Research Participation Program at the Centers for Disease Control and Prevention, administered by the Oak Ridge Institute for Science and Education through an interagency agreement between the U.S. Department of Energy and CDC. We thank Audrey Matheny, Justin S. Lee, Dakota T. Howard, Whitni Davidson, and Shatavia Morrison. The findings and conclusions in this report are those of the author(s) and do not necessarily represent the views of the Centers for Disease Control and Prevention.

## Author contributions

C.M.G. and Y.L.: conception and design; C.M.G., D.M., S.S., H.Z., D.B., R.M., J.P., A.M., M.L., S.G., M.B., D.H., G.R., J.H., A.B., M.S., T.K., V.W., K.W.: data acquisition; C.M.G., J.T., M.P., D.M., D.B., E.E.H., A.M., W.D., J.S.L., D.H., S.M., K.O., J.L.R., L.K., M.H.S., M.R.W., P.S.S., A.M.M.: data analysis; C.M.G., Y.L., C.L.H.: drafted the manuscript. All authors reviewed and revised the final manuscript.

## Competing interests

The authors declare no competing interests
