## [Peer Review file · Nature Communications]

Patterns of genomic deletions in monkeypox virus during the global 2022 mpox outbreak in the United States

Corresponding Author: Dr Crystal Gigante

Version 0:

Reviewer comments:

Reviewer #1

(Remarks to the Author)

The manuscript is well constructed and written. It is a descriptive study examining distinct deletions present in MPXV genomes, specific to the state of Louisiana.

Comments

- (1) The authors described the deletions observed in the MPXV genomes in Louisiana. It would be helpful to understand how these deletions were confirmed both from an experimental and bioinformatics point of view. Although metagenomics approach is considered “unbiased”, dropout in sequencing as a result of low sequencing depth can occur. Information such as reads, depth, viral load would be helpful in painting the overall picture.
- (2) What is the limit of detection for identifying deletions using the current methodology? Can low level deletions be missed?
- (3) The bioinformatics analysis for deletions need to be described more thoroughly, including the process of identifying these mutations. Are individual deletions inspected manually? Are there risks of missing smaller deletions?
- (4) The authors mentioned about genomic changes, including duplications, recombinations, horizontal gene transfer (line 76-77). Please explain why the study only looks at deletions?
- (5) Functional assessment to determine the impact of deletions would be useful. Example, the deletions in the target region of clade II mpox diagnostic PCR test was mentioned (line 274). Is it anticipated that there will be a total loss of PCR sensitivity or partial? Has any functional work been done to confirm the impact on diagnostic assay?
- (6) Can the author comment on whether the deletions are likely to be deleterious or adaptive, and whether they are likely to result in increased transmissibility?
- (7) Deletions in global context would be helpful to understand if they are present in other regions of the world. The authors comment about “unique deletions” but there is no mention in the methodology/analysis on the comparison with global sequences.

Minor:

- (1) Could authors please specify which nextclade dataset are used to perform lineage calling? This is updated regularly and the most recent dataset should be used.
- (2) Please specify which sequences were used in the phylo tree analysis. Are these global public sequences or only restricted to the US?

Reviewer #2

(Remarks to the Author)

The authors investigate the patterns of large genomic deletions in Clade IIb lineage B.1 Mpox virus (MPXV) genomes during routine surveillance at the CDC in the U.S. during 2022 outbreak. Twenty-nine (2,6%) unique deletions greater than 500 bp were identified in 2,362 MPXV genome sequences. These deletions were primarily located towards the terminal ends of the MPXV genome and resulted in extensive predicted gene loss as well as several novel predicted gene products. Each deletion mutant was identified as the major sequence present, suggesting that the virus was replicative. Most large deletions were rare. However, a 3,370 bp deletion was prevalent in isolates from Louisiana state, and a 913 bp deletion appeared in multiple sub-lineages across several countries, suggesting independent occurrences. The prevalence of genomic deletions in poxviruses is not well understood. Some studies have reported on SNP changes in MPXV sequences. However, a few

studies like this one describe large deletions during genomic surveillance. The study highlights the role of large deletions in poxvirus evolution and their potential impact on diagnostics and therapeutics. The findings underscore the importance of continued genomic surveillance to monitor viral changes, inform public health strategies, and support the use of diagnostic tests or medical countermeasures that target multiple MPXV proteins.

Some concerns should be addressed before acceptance of this manuscript:

1. Line 1: The author should add: during the global 2022 outbreak in the US. Also, may change monkeypox virus for MPXV in all the text.
2. Line 74. Add a space in approximately 6
3. Figure 1. The Read depth coverage from metagenomic sequencing of MPXV genomes from the upper sample with the 14,410 deletion is below 100, which may hamper the accurate identification of this deletion.
4. It would be clearer for the reader to place in Figure 1 a schematic of the predicted architecture of MPXV proteins disrupted in the genomes with the 3,370 bp and 913 bp deletions.
5. In Table 1 I recommend expanding the information about the type and the function (predicted) of proteins.
6. Line 148: In general, deletions were located towards the terminal ends of the MPXV genome (Figure 2). Except for one 640 bp deletion (MPXVdel 3). The authors may add more information concerning this deletion.
7. Figure 2. The graphic representation of the deletion is missing in MPXVdel 28.
8. Lines 217-219: The 28 MPXV sequences containing this deletion were generated from samples from 29 cases from Louisiana and 1 each from Utah, Texas, and Georgia. There is an error in the number of sequences or samples because they are different (28 vs. 32).
9. Lines 222-224: Phylogenetic analysis revealed the MPXV sequences containing the 3,370 bp deletion formed a monophyletic clade, separated from other MPXV sequences from Louisiana, suggesting a common ancestor (Figure 3B, Figure S3B). To which sublineages of B.1 do these sequences belong?
10. Line 229 and 238: Add LA in parentheses after Louisiana.
11. Line 253: place the number 2 in the first subline as shown below: including B.1.2, B.1.3, B.1.9, and B.1.15
12. It would be interesting to know if the sequences with deletions also contain insertions and nonsense mutations.
13. Lines 322-325: Add the references of this previous research.
14. Line 426: Deletion was confirmed by examination of raw read sequences at the boundaries of the suspected deletion. This should be discussed in more detail in the manuscript. The authors may consider Sanger sequencing of this region to confirm the deletion.
15. What was the source of viral DNA? And, what was the Ct value selected for sequencing?
16. The authors should mention the Bioethical approval for this study.
17. Are there differences in the Ct between the samples with genome deletions and without deletions?
18. Are there differences between patients with MPOXV genome deletions and those without deletions (immunocompromised or immunocompetent, community transmission or travel-related case, and sample type)?
19. Figure S2 is not easy to understand: it should be presented more clearly.

Version 1:

Reviewer comments:

Reviewer #1

(Remarks to the Author)

The authors have adequately addressed all my comments.

Reviewer #2

(Remarks to the Author)

The authors satisfactorily addressed most of the concerns. However, there are still some minor corrections that should be addressed before the final acceptance of this manuscript:

- 1) In Table 1, most of the symbols in the legend are not in the table, and those that are there do not match the information in the legend. Furthermore, the table is formatted larger than the page size, resulting in some information being lost.
- 2) On line 177, the authors examined the predicted impact of each of the 31 unique deletions on predicted genes, but then, they only mention the 29 from the previous paper version. The authors should adjust that paragraph to the 31 unique deletions now reported.
- 3) Figure 3: Line 246: Now the branches are orange, not red. Line 249: include the state of Louisiana (LA: Louisiana) or keep Louisiana (LA) on line 245.
- 4) In lines 333-335, the references to that previous work are still missing. Line 334: replace OPXVs with MPXVs

REVIEWER COMMENTS

Reviewer #1 (Remarks to the Author):

The manuscript is well constructed and written. It is a descriptive study examining distinct deletions present in MPXV genomes, specific to the state of Louisiana.

Comments

(1) The authors described the deletions observed in the MPXV genomes in Louisiana. It would be helpful to understand how these deletions were confirmed both from an experimental and bioinformatics point of view. Although metagenomics approach is considered “unbiased”, dropout in sequencing as a result of low sequencing depth can occur. Information such as reads, depth, viral load would be helpful in painting the overall picture.

Thank you. To avoid erroneously including coverage dropout in our counts of genomic deletions, we required at least 10 reads to span the deletion junction to be included as a genomic deletion in this study. We have added Text S1 in this resubmission to provide additional details on how deletions were identified and verified.

(2) What is the limit of detection for identifying deletions using the current methodology? Can low level deletions be missed?

Yes, as mentioned above, we required at least 10 reads to span the deletion region in our study. Some samples with low or uneven sequence coverage may contain deletions that will be missed. Some high coverage sample may contain minority viruses with deletions that would also be missed. We do not attempt to determine a limit of detection for this method. We have included representative images of read mapping profiles from samples producing overall low read depth to show how (in some cases) deletions can still be easily observed with low overall read depth (Figure S5).

(3) The bioinformatics analysis for deletions need to be described more thoroughly, including the process of identifying these mutations. Are individual deletions inspected manually? Are there risks of missing smaller deletions?

We have added Text S1 in this resubmission to provide additional details on how deletions were identified and verified. In this paper, we only describe deletions >500 bp since they were manually identified by eye based on read mapping profiles. Smaller

deletions do exist and are relatively common in MPXV, especially in low complexity regions, but they are not discussed here.

(4) The authors mentioned about genomic changes, including duplications, recombinations, horizontal gene transfer (line 76-77). Please explain why the study only looks at deletions?

Our original preprint

(<https://www.biorxiv.org/content/10.1101/2022.09.16.508251v1.abstract>) did include discussion of other genomic rearrangements; however, we decided to split them into separate manuscripts as the studies progressed and stories diverged.

(5) Functional assessment to determine the impact of deletions would be useful. Example, the deletions in the target region of clade II mpox diagnostic PCR test was mentioned (line 274). Is it anticipated that there will be a total loss of PCR sensitivity or partial? Has any functional work been done to confirm the impact on diagnostic assay?

This work has been published in reference 38: Garrigues JM, *et al.* Identification of Human Monkeypox Virus Genome Deletions That Impact Diagnostic Assays. *J Clin Microbiol* **60**, e0165522 (2022). There is total loss of PCR sensitivity (samples not detected, primer and probe binding regions are deleted). We have added clarification to the sentence.

(6) Can the author comment on whether the deletions are likely to be deleterious or adaptive, and whether they are likely to result in increased transmissibility?

Our delayed and retrospective attempts to follow up on clinical features of these cases did not reveal any evidence of more severe disease or other distinguishing clinical features in these cases; however, information was very limited. Thus, we prefer to remain conservative in our speculation on what role the deletions may have on individual viruses.

(7) Deletions in global context would be helpful to understand if they are present in other regions of the world. The authors comment about “unique deletions” but there is no mention in the methodology/analysis on the comparison with global sequences.

Comparison with global sequences is challenging, as confirmation of deletions relies on careful analysis of raw read data. Wide use of enrichment protocols makes it much more difficult to identify deletions, as read depths show much higher variability and drop out using either PCR or probe-based methods in our hands. We hope surveillance of

deletions will be improved with increase awareness from others reading our findings and reviewing our data, but this study focuses on analysis of data we generated according to the methods described.

Minor:

(1) Could authors please specify which nextclade dataset are used to perform lineage calling? This is updated regularly and the most recent dataset should be used.

This information has been added to the methods. We re-ran all sequences in Figure 3 and 4 using a current reference dataset. Since all data are from 2022, 2022 - 2023 naming for lineages is used, and sequences from E, F lineages were not included.

(2) Please specify which sequences were used in the phylotree analysis. Are these global public sequences or only restricted to the US?

The phylogeny in Figure 3 includes only MPXV sequences from the US state of Louisiana and sequences from other states that shared the same or a similar deletion. This has been clarified in the figure legend. Accession or isolate numbers have been added to Figure 3 to make it easier to interpret. The phylogeny in Figure 4 contains US and non-US sequences. All accession numbers from the phylogenetic trees are included in the corresponding supplemental figure file (S4).

Reviewer #2 (Remarks to the Author):

The authors investigate the patterns of large genomic deletions in Clade IIb lineage B.1 Mpox virus (MPXV) genomes during routine surveillance at the CDC in the U.S. during 2022 outbreak. Twenty-nine (2,6%) unique deletions greater than 500 bp were identified in 2,362 MPXV genome sequences. These deletions were primarily located towards the terminal ends of the MPXV genome and resulted in extensive predicted gene loss as well as several novel predicted gene products. Each deletion mutant was identified as the major sequence present, suggesting that the virus was replicative. Most large deletions were rare. However, a 3,370 bp deletion was prevalent in isolates from Louisiana state, and a 913 bp deletion appeared in multiple sub-lineages across several countries, suggesting independent occurrences. The prevalence of genomic deletions in poxviruses is not well understood. Some studies have reported on SNP changes in MPXV sequences. However, a few studies like this one describe large deletions during genomic surveillance. The study highlights the role of large deletions in poxvirus evolution and their potential impact on diagnostics and therapeutics. The findings underscore the importance of continued genomic surveillance to monitor viral changes,

inform public health strategies, and support the use of diagnostic tests or medical countermeasures that target multiple MPXV proteins.

Some concerns should be addressed before acceptance of this manuscript:

1. Line 1: The author should add: during the global 2022 outbreak in the US. Also, may change monkeypox virus for MPXV in all the text.

The title has been changed as suggested and the abbreviation MPXV is now “monkeypox virus (MPXV) in the abstract.

2. Line 74. Add a space in approximately6

This has been corrected.

3. Figure 1. The Read depth coverage from metagenomic sequencing of MPXV genomes from the upper sample with the 14,410 deletion is below 100, which may hamper the accurate identification of this deletion.

We have amended this figure to include two examples with high coverage and have moved examples of low coverage deletions to the Figure S5. We have also added Text S1 to better describe how deletion mutations were identified and confirmed.

4. It would be clearer for the reader to place in Figure 1 a schematic of the predicted architecture of MPXV proteins disrupted in the genomes with the 3,370 bp and 913 bp deletions.

This information was included in the original Figure S2 for the 3,370 bp deletion. We have completely replaced Figure S2 with Figure S3 to include a better visualization of the predicted impacts of these two deletions.

5. In Table 1 I recommend expanding the information about the type and the function(predicted) of proteins.

A few additional details have been added to Table 1 for some of the smaller deletions. However, for ease of viewing Table 1, readers are directed to Table S1, which has much more detailed information.

6. Line 148: In general, deletions were located towards the terminal ends of the MPXV genome (Figure 2). Except for one 640 bp deletion (MPXVdel 3). The authors may add more information concerning this deletion.

Additional information has been added in the text.

7. Figure 2. The graphic representation of the deletion is missing in MPXVdel 28.

Thank you, this has been corrected.

8. Lines 217-219: The 28 MPXV sequences containing this deletion were generated from samples from 29 cases from Louisiana and 1 each from Utah, Texas, and Georgia. There is an error in the number of sequences or samples because they are different (28 vs. 32).

Thank you, this has been corrected. There were two unique deletions from LA that are now included as separate deletions in Table 1 / Figure 2 and one low coverage sequence with the 3,370 bp deletion that has been added to the final dataset.

9. Lines 222-224: Phylogenetic analysis revealed the MPXV sequences containing the 3,370 bp deletion formed a monophyletic clade, separated from other MPXV sequences from Louisiana, suggesting a common ancestor (Figure 3B, Figure S3B). To which sublineages of B.1 do these sequences belong?

This information has been added.

10. Line 229 and 238: Add LA in parentheses after Louisiana.

This has been corrected.

11. Line 253: place the number 2 in the first subline as shown below: including B.1.2, B.1.3, B.1.9, and B.1.15

Thank you, this has been corrected.

12. It would be interesting to know if the sequences with deletions also contain insertions and nonsense mutations.

The MPXV viruses described here do not contain genomic insertions or duplications outside of low complexity areas. Viruses with evidence of large genomic rearrangements or other large genomic changes other than simple deletions were excluded from the dataset (Line 116). The data are soon to be available publicly, so others are free to investigate for mutations in genes of interest.

13. Lines 322-325: Add the references of this previous research.

References have been added.

14. Linea 426: Deletion was confirmed by examination of raw read sequences at the boundaries of the suspected deletion. This should be discussed in more detail in the manuscript. The authors may consider Sanger sequencing of this region to confirm the deletion.

Supplemental text 1 was added to provide more detail into how suspect deletions were identified and how deletions were confirmed.

15. What was the source of viral DNA? And, what was the Ct value selected for sequencing?

16. The authors should mention the Bioethical approval for this study.

This information has been added to the methods.

17. Are there differences in the Ct between the samples with genome deletions and without deletions?

We compared Ct values of the deletion dataset to 5,130 samples tested at CDC (including samples with genomic deletions). For the entire dataset, the average Ct value using clade II-specific test among samples with Ct <29.5 was 23.4 (ranging from 15.1 to 29.4). Among the deletion dataset (n=73), the average Ct value was 22.6 after removing two samples that failed to amplify in the clade II-specific PCR test (ranging from 16.5 to 29.79). We have added Ct values to Table S2, but we prefer not to add this analysis into the text because we feel our study was biased towards samples with low Ct (Ct <29) and is not well designed to answer this question.

18. Are there differences between patients with MPOXV genome deletions and those without deletions (immunocompromised or immunocompetent, community transmission or travel-related case, and sample type)?

We did not observe any obvious differences; however, the delayed and retrospective nature of our inquiries into these cases yielded limited responses. With only 1 – 2 cases for most deletions, we refrain from making any conclusions about those cases. For the 3,370 bp deletion, no clinical or other differences were observed (other than geographic clustering) compared to other clade IIb cases in the USA.

19. Figure S2 is not easy to understand: it should be presented more clearly.

This figure has been remade as suggested as Figure S3 and now includes the protein comparison for 3,370 bp deletion predicted product and compare the 913 bp deletion to OPG023 locus in other orthopoxviruses.

Editorial comments:

POLICIES AND FORMS REQUIRED FOR RESUBMISSION

Editorial policy checklist:

<https://www.nature.com/documents/nr-editorial-policy-checklist.pdf>

Reporting summary:

These checklists have been completed.

DATA AND CODE AVAILABILITY

* All Nature Communications manuscripts must include a “Data Availability” section after the Methods section but before the References. If any of the data can only be shared on request or are subject to restrictions, please specify the reasons and explain how, when, and by whom the data can be accessed. For more information on this policy and a list of examples, see:

<https://www.nature.com/documents/nr-data-availability-statements-data-citations.pdf>

* Proteomic datasets must be deposited in a publicly accessible database, and accession codes and associated hyperlinks must be provided in the “Data Availability” section.

* All DNA sequencing, RNA sequencing or microarray data must be deposited in an approved, publicly accessible repository listed here (<https://www.nature.com/nature-research/editorial-policies/reporting-standards#availability-of-data>), and relevant accession codes should be stated in the data availability statement.

Accession numbers have been added to the manuscript.

* Please replace your bar graphs with plots that feature information about the distribution of the underlying data. All data points should be shown for plots with a sample size less than 10. For larger sample sizes, please consider box-and-whisker or violin plots as alternatives. Measures of centrality, dispersion and/or error bars should be plotted and described in the figure legend.

We have added sample size of each group to the bar plot, which we feel meets the requirement of featuring information about the distribution of the underlying data. We would like to continue to display the proportion of the MPXV genomes with the deletion among samples submitted to CDC over time. We hope this change has addressed this point.

ORCID

* Nature Communications is committed to improving transparency in authorship. As part of our efforts in this direction, we are now requesting that all authors identified as 'corresponding author' create and link their Open Researcher and Contributor Identifier (ORCID) with their account on the Manuscript Tracking System prior to acceptance. ORCID helps the scientific community achieve unambiguous attribution of all scholarly contributions.

ORCID added: 0000-0002-2169-7156

AUTHOR CHANGES ON REVISION

If there are any changes to the author list in the revised manuscript, please use this approval form www.nature.com/documents/nr-author-list-change-form.pdf, arranging for all authors on your paper to sign the statement confirming that they agree to the author list being changed, and add this document to your resubmission.

Not applicable

Reviewer #1 (Remarks to the Author):

The authors have adequately addressed all my comments.

We thank the reviewer for their time.

Reviewer #2 (Remarks to the Author):

The authors satisfactorily addressed most of the concerns. However, there are still some minor corrections that should be addressed before the final acceptance of this manuscript:

We thank the reviewer for their time and have worked to address the remaining comments below.

1) In Table 1, most of the symbols in the legend are not in the table, and those that are there do not match the information in the legend. Furthermore, the table is formatted larger than the page size, resulting in some information being lost.

Thank you, the table size has been adjusted, and the symbols have been added back.

2) On line 177, the authors examined the predicted impact of each of the 31 unique deletions on predicted genes, but then, they only mention the 29 from the previous paper version. The authors should adjust that paragraph to the 31 unique deletions now reported.

Thank you, this has been corrected.

3) Figure 3: Line 246: Now the branches are orange, not red. Line 249: include the state of Louisiana (LA: Luisiana) or keep Louisiana (LA) on line 245.

Thank you, this has been corrected.

4) In lines 333-335, the references to that previous work are still missing. Line 334: replace OPXVs with MPXVs

Thank you, references and a clarification statement have been added to that sentence.